# Dinactin: A New Antitumor Antibiotic with Cell Cycle Progression and Cancer Stemness Inhibiting Activities in Lung Cancer

**DOI:** 10.3390/antibiotics11121845

**Published:** 2022-12-19

**Authors:** Anchalee Rawangkan, Pattama Wongsirisin, Grissana Pook-In, Achiraya Siriphap, Atchariya Yosboonruang, Anong Kiddee, Jureeporn Chuerduangphui, Nanthawan Reukngam, Acharaporn Duangjai, Surasak Saokaew, Ratsada Praphasawat

**Affiliations:** 1Division of Microbiology and Parasitology, School of Medical Sciences, University of Phayao, Phayao 56000, Thailand; 2UNIt of Excellence on Clinical Outcomes Research and IntegratioN (UNICORN), School of Pharmaceutical Sciences, University of Phayao, Phayao 56000, Thailand; 3Department of Medical Services, National Cancer Institute, Bangkok 10400, Thailand; 4Department of Microbiology, Faculty of Science, Kasetsart University, Bangkok 10900, Thailand; 5Laboratory of Organic Synthesis, Chulabhorn Research Institute, Bangkok 10210, Thailand; 6Division of Physiology, School of Medical Sciences, University of Phayao, Phayao 56000, Thailand; 7Division of Social and Administrative Pharmacy, Department of Pharmaceutical Care, School of Pharmaceutical Sciences, University of Phayao, Phayao 56000, Thailand; 8Department of Pathology, School of Medicine, University of Phayao, Phayao 56000, Thailand

**Keywords:** antitumor antibiotic, cell cycle arrest, CSC, dinactin, NSCLC, stemness

## Abstract

Lung cancer, especially non-small cell lung cancer (NSCLC), is one of the most complex diseases, despite the existence of effective treatments such as chemotherapy and immunotherapy. Since cancer stem cells (CSCs) are responsible for chemo- and radio-resistance, metastasis, and cancer recurrence, finding new therapeutic targets for CSCs is critical. Dinactin is a natural secondary metabolite produced by microorganisms. Recently, dinactin has been revealed as a promising antitumor antibiotic via various mechanisms. However, the evidence relating to cell cycle progression regulation is constrained, and effects on cancer stemness have not been elucidated. Therefore, the aim of this study is to evaluate the new function of dinactin in anti-NSCLC proliferation, focusing on cell cycle progression and cancer stemness properties in Lu99 and A549 cells. Flow cytometry and immunoblotting analyses revealed that 0.1–1 µM of dinactin suppresses cell growth through induction of the G_0_/G_1_ phase associated with down-regulation of cyclins A, B, and D3, and cdk2 protein expression. The tumor-sphere forming capacity was used to assess the effect of dinactin on the cancer stemness potential in NSCLC cells. At a concentration of 1 nM, dinactin reduced both the number and size of the tumor-spheres. The quantitative RT-PCR analyses indicated that dinactin suppressed sphere formation by significantly reducing expression of CSC markers (i.e., *ALDH1A1*, *Nanog*, *Oct4*, and *Sox2*) in Lu99 cells. Consequently, dinactin could be a promising strategy for NSCLC therapy targeting CSCs.

## 1. Introduction

Lung cancer remains the malignant tumor with the highest mortality, accounting for 18% of all cancer deaths worldwide [1]; most of these are non–small cell lung cancer (NSCLC) [2,3]. Lung cancer is also the most common type of cancer in Thailand, accounting for 14.1% of all cancers and having the second-highest mortality rate (18.7%) after liver cancer (20.3%) [4]. With the advancement of molecular targeted therapy and immunotherapy, the treatment strategy of NSCLC has shifted toward precision management. However, identifying patients who respond to immune checkpoint inhibitors or who eventually do not respond remains a significant challenge [5], and treatments for NSCLC with immune checkpoint inhibitors are increasing healthcare costs [6,7]. Therefore, exploring the molecular mechanisms of natural bioactive compounds with anti-cancer activity is gaining tremendous interest in the field of oncology [8,9]. NSCLC is prone to therapeutic resistance and tumor recurrence. Self-renewing of cancer stem cells (CSCs) was demonstrated to exert a key role in the intrinsic resistance to chemotherapy and radiotherapy [10,11]. The development of anti-cancer agents targeting CSCs is needed to improve the efficacy of cancer treatment and to prevent relapse in NSCLC patients.

Dinactin (Appendix A) belongs to the macrotetrolide family, which has a wide range of biological activities, including antimicrobial, insecticidal, acaricidal, antiprotozoal, antiparasitic, and immunosuppressive properties [12,13]. Dinactin is a secondary metabolite produced by various *Streptomyces* species, such as *S. globisporus* [14], *S. araujoniae* [15], *S. puniceus* [16,17], and *S. cavourensis* [18], as well as *S. badius* [19]. A hallmark of dinactin is that it is easily purified and can be obtained in low-cost media cultures, which yield up to 160.8 mg/L [20]. The mechanism of action of dinactin on antibiotic activity is mediated by damaged bacterial membranes, which result in the release of bioactive compounds from the bacteria. Dinactin also stimulates mitochondrial ATPase activity and causes rapid hydrolysis of ATP, resulting in mitochondrial dysfunction in bacteria. Dinactin has additional functions as a monovalent cation ionophore with a high selectivity for potassium and ammonium [12]. On the other hand, some studies report that dinactin is a new promising antitumor antibiotic through several mechanisms: inhibition of cellular proliferation in various cancer cells, induction of cell cycle arrest and apoptosis, reduction of clonogenic survival, inhibition of cell migration and invasion, and blocking the Wnt/β-catenin signaling pathway [16,17,21]. This suggests that dinactin could be a multi-functional drug that acts through multiple signaling pathways.

Recent studies have revealed that treating cancer with antibiotics is an innovative strategy for eradicating CSCs. FDA-approved antibiotics such as azithromycin, doxycycline, tigecycline, pyrvinium pamoate, and chloramphenicol inhibit tumor-sphere formation in various cancer cells while being non-toxic to normal cells by associating with mitochondrial dysfunction [22,23].

All of this leads us to hypothesize that dinactin might have an inhibiting activity on cancer stemness in NSCLC cells. Therefore, in this study, we assessed the anti-cancer activity of dinactin in NSCLC cells (Lu99 and A549 cells) by investigating cell proliferation and cancer stemness properties. Our findings revealed that dinactin specifically inhibited cell proliferation by cell cycle arrest at the G_0_/G_1_ phase and reduced the number and size of the tumor spheres, indicating that dinactin inhibited cancer stemness in NSCLC cells. Hence, dinactin is a possible antitumoral for NSCLC therapy targeting CSCs.

## 2. Results

### 2.1. Dinactin Inhibited Cancer Cell Growth by Inducing G_0_/G_1_ Arrest

The effects of dinactin on growth of Lu99 and A549 cells were measured by the MTT assay. Cells were treated with dinactin at concentrations of 0, 0.1, and 1 μM for 1 to 5 days. As shown in Figure 1a, dinactin at 0.1 µM strongly inhibited cell growth within 24 h, similar to higher concentration and long-term treatment (1 µM for 5 days). The half-maximal inhibitory concentration (IC_50_) for dinactin in Lu99 and A549 cells were 2.06 ± 0.21 nM and 3.26 ± 0.16 nM, respectively. However, the IC_50_ value in normal fibroblast cells was more than a 100-fold higher concentration (Appendix A). These findings indicate that dinactin acts as a cancer-specific molecule and inhibits NSCLC cell growth at low concentrations.

To investigate the mechanism of how dinactin inhibits cellular proliferation, we evaluated its effect on cell cycle progression. Lu99 and A549 cells were treated with 0.1 and 1 µM of dinactin at 48 h, and the percentage of cell cycle distribution in the G_0_/G_1_, S, and G_2_/M phases was analyzed using flow cytometry. Metformin, a type 2 diabetes treatment agent, was used as a positive control for cell cycle arrest at the G_0_/G_1_ phase at a concentration of 24.5 mM [22]. We found that dinactin induced cell cycle arrest at the G_0_/G_1_ phase and moderates the cell cycle checkpoint similar to metformin.

As shown in Figure 1b, treatment with 0.1 and 1 µM of dinactin significantly increased the percentage of cells in the G_0_/G_1_ phase from 64.57 ± 1.46% (control) to 74.4 ± 3.96% (0.1 µM) and 72.97 ± 2.15% (1 µM) in Lu99 cells. That of metformin-treated cells accounted for 89.1 ± 1.40% as a positive control. Dinactin also reduced the percentage of cells in the G_2_/M phase from 21.6 ± 1.61% to 12.36 ± 5.37% (0.1 µM), 14.07 ± 2.42% (1 µM), and 7.01 ± 0.5% (metformin), indicating that dinactin significantly delays progression of the cell cycle into G_2_/M phase. There was no discernible difference in the S phase. Similar results were found in A549 cells (Figure 1c). The percentages of cells in the G_0_/G_1_ phase were 37.50 ± 0.71% for the control, 54.20 ± 5.73% and 53.97 ± 7.77% for 0.1 and 1 µM of dinactin, respectively, and 74.97 ± 4.53% for metformin, whereas there was no discernible change in the S and G_2_/M phases. These results suggest that dinactin suppresses the cell growth of Lu99 and A549 through induction of the G_0_/G_1_ phase of cell cycle arrest.

### 2.2. Dinactin Down-Regulated Cyclins A, B, and D3, cdk2 and PCNA in Lu99 and A549 Cells

To understand the mechanism of dinactin for inducing G_0_/G_1_ arrest, we next examined the expression of proteins regulating cell cycle progression after treatment with 0.1 and 1 µM of dinactin for 24 and 48 h in Lu99 and A549 cells. The protein levels of cyclins A, B, and D3, cdk2, proliferating cell nuclear antigen (PCNA), and p21 protein, which were associated with promoting cell cycle progression from the G_1_ to the S phase, were examined by western blot analyses. As shown in Figure 2a, in Lu99 cells, treatment with dinactin at 0.1 and 1 µM significantly reduced the expression of cyclin A, cyclin B, cyclin D3, cdk2, and PCNA, whereas the level of p21, a CDK inhibitor, did not. Similar results were found in A549 cells (Figure 2b). Furthermore, dinactin seems to be more effective after 48 h than after 24 h of treatment. It is important to note that p21 protein, a regulator of the cell cycle, DNA replication, and apoptosis, did not significantly change in both Lu99 and A549 cells. These results support the finding that dinactin suppresses cell growth by inducing G_0_/G_1_ phase cell cycle arrest via the downregulation of cyclins A, B, and D3, cdk2, and PCNA, resulting in inhibition of cancer cell proliferation.

### 2.3. Dinactin Inhibited Tumor-Sphere Formation in Lu99 and A549 Cells

Next, we investigated the inhibitory effects of dinactin on cancer stemness properties by tumor-sphere formation assay. Lu99 and A549 cells were cultured under non-attached and serum-free CSCs culture medium in the presence of 0.1 and 1 nM of dinactin for 14 days. The number of tumor-spheres greater than 100 µm was counted and kept for RNA extraction. As shown in Figure 3a,b, non-treated Lu99 cells produced 195.5 ± 45.3 tumor-spheres (control), whereas treatment with 0.1 and 1 nM of dinactin inhibited tumor-sphere formation by 195.3 ± 39.3 and 9.33 ± 11.3 (95.2% inhibition), respectively. In A549 cells, similar results were obtained: treatment with 0.1 and 1 nM of dinactin inhibited tumor-sphere formation from 144.5 ± 69.5 to 109.7 ±15.9 (24.1% inhibition) and 41.0 ± 15.8 (71.6% inhibition), respectively. Furthermore, dinactin reduced the size of tumor-spheres, as shown in Table 1. At a concentration of 1 nM of dinactin, the average size of tumor-spheres was reduced from 175.54 ± 55.78 µm to 128.82 ± 20.07 µm (26.6% inhibition) in Lu99 cells and from 158.72 ± 47.40 µm to 134.87 ± 25.71 µm (15.0% inhibition) in A549 cells. These are new findings that highlight the very low concentration of dinactin (1 nM) capable of inhibiting CSC in NSCLC cells.

### 2.4. Effects of Dinactin on Stemness Related Gene Expressions in Lu99 and A549 Tumor-Spheres

To understand how dinactin inhibits CSCs properties in the lung cancer cells, we next investigated expression of genes of the well-known lung CSC markers, including *ALDH1A1*, *Nanog*, *CD133*, *Oct4*, and *Sox2* genes by qRT-PCR. Tumor-spheres in Lu99 cells expressed *ALDH1A1*, *Nanog*, *Oct4*, and *Sox2* genes, but not *CD133*. Dinactin significantly reduced the relative expression of these four genes in Lu99 cells in a dose-dependent manner (Figure 4a). On the other hand, tumor-sphere in A549 cells expressed only three genes (*ALDH1A1*, *Oct4*, and *Sox2*), and dinactin significantly reduced the relative expression of only *ALDH1A1* (Figure 4b). Although the difference in mechanisms between Lu99 and A549 cells is not clear, this is the first report of a new dinactin function on cancer stemness properties.

## 3. Discussion

The primary goal of an antibiotic is to prevent and treat infectious bacterial diseases through four major mechanisms of action: inhibition of bacterial cell wall synthesis, increased permeability of bacterial cell membranes, interference with bacterial protein synthesis, and inhibition of bacterial nucleic acid replication and transcription [24]. Furthermore, antibiotics such as salinomycin, ciprofloxacin, gemifloxacin, doxorubicin, bleomycin, enediynes, and mitomycin have anti-tumor properties [25,26,27]. Evidence of antitumor antibiotics on the mode of anticancer activity has been reported via various mechanisms—anti-proliferative [28], suppression of self-renewal abilities [29,30,31], inhibition of autophagy [32], induced apoptosis [33,34], anti-Epithelial-Mesenchymal-Transition (EMT) [35], cell-cycle checkpoint control [36], and inducing mitochondria dysfunction and oxidative stress in various cancer cells [37].

In this study, we emphasized that dinactin is a new antitumor antibiotic that inhibits the activities of cell cycle progression and cancer stemness in lung cancer. It is important to note that IC_50_ values of dinactin in NSCLC were nanomolar concentration (IC_50_ = 2.06 ± 0.21 nM for Lu99 and IC_50_ = 3.26 ± 0.16 nM for A549), which are much lower than previous reports for lung, colon, breast, and liver cancer cell lines; IC_50_ values were from 1.1 to 9.7 µM [16]. Based on these findings, we investigated the mechanism of action of dinactin in the regulation of cell cycle progression, a well-known cell proliferation mechanism.

According to a previous report, dinactin inhibits G_1_/S progression and decreases cyclin D1 expression in HCT-116 cells [16]. We encountered similar activity in NSCLC, where dinactin downregulated the cyclins A, B, D3, cdk2, and PCNA, a complex cascade of cellular events, but did not induce a significant change for the p21. Since we did not observe a sub-G1 peak, an indicator of apoptotic cell death [38,39], this new finding suggests that the anti-proliferative effect of dinactin might not be related to the induction of cell death. Indeed, p21 has been shown to cause cell cycle arrest in the G_0_/G_1_ phase by inhibiting the activity of the CDK/cyclin D complex, as well as to protect against apoptosis in response to other stimuli [36,40].

Self-renewing lung cancer stem cells have been shown to play a key role in tumor initiation and progression, and they also contribute to drug resistance, tumor recurrence, and metastasis. This is the first report indicating that dinactin suppresses the cancer stemness properties of NSCLC cells at 1 nM concentration. In Lu99 cells, dinactin significantly downregulated *ALDH1A1*, *Nanog*, *Oct4*, and *Sox2* mRNA expression, which are transcription factors that are required to maintain pluripotency in lung cancer. Although dinactin significantly inhibited tumor-sphere formation in A549 cells, stemness gene expression did not change distinctly. Since Lu99 and A549 cell lines have several differences in malignancy, metastatic potential, and histological types (Lu99 is a lung large cell carcinoma with highly metastatic and A549 is a lung adenocarcinoma), these differences attributed to differences of effects on stemness genes [41]. However, future studies need to investigate the anti-metastatic potential of dinactin. Moreover, the effectiveness and mode of mechanism(s) of dinactin on CSCs inhibitors must be investigated further for future in vivo and clinical studies.

In this study, we focused on *ALDH1A1*, *CD133*, *Nanog*, *Oct4*, and *Sox2*, which are transcription factors required for lung cancer pluripotency. Indeed, *ALDH1A1* controls cell cycle checkpoints by regulating KLF4 and p21 proteins, which contribute to apoptosis inhibition in cancer stem cells [42,43]. *CD133* expression levels are linked to cell cycle DNA profiles, and *CD133* deficiency reduces cell proliferation significantly. A comparison of wild-type and knockout human embryonic stem cells showed a significantly decreased population in the S phase, whereas the cell population in the G_1_ phase was increased [44,45]. *Nanog* regulates S-phase entry in human embryonic stem cells via transcriptional regulation of cell cycle regulatory components, resulting in an increase in pluripotency and cell growth [46]. *Oct4* is a cell cycle promoter that removes the inhibitor p21 of cell cycle progression in the G_1_ phase and stimulates entry into the S phase, leading to promoted proliferation and cell cycle progression [47]. *Sox2* regulates the cell cycle by interacting with direct and indirect cyclin D as well as cycle inhibitors p21 and p27, resulting in proliferative stem cells [48]. Therefore, treatment with dinactin does indeed suppress the stemness property that is related to cell cycle progression in lung cancer cells.

Salinomycin, a similar membrane ionophore antibiotic to dinactin, has been identified as an antitumor antibiotic for several types of CSC treatments, including lung, liver, breast, and ovarian cancer [49,50]. The mode of action of salinomycin as a CSC inhibitor involves various mechanisms, such as attenuating liver cancer stem cell motility by enhancing cell stiffness and increasing F-actin formation via the FAK-ERK1/2 signaling pathway [31]. The combination of salinomycin and metformin effectively inhibits the formation of spheres in NSCLC cells with varying EGFR, KRAS, EML4/ALK, and LKB1 status [29]. In addition, treatment with salinomycin and paclitaxel decreased the viability of ovarian CSCs and promoted cell apoptosis [30]. Based on our results that dinactin showed similar effects to metformin, we expect that a combination of dinactin with salinomycin or conventional anti-cancer drugs can improve anti-cancer activity in lung cancer treatment.

It is important to note that using antibiotics for treatment of cancer carries a double-edged sword because it may also induce cancer generation by disrupting intestinal microbiota and causing microbial imbalance, which further promotes chronic inflammation, alters normal tissue metabolism, leads to genotoxicity, and weakens the immune response to bacterial malnutrition, thereby adversely impacting cancer treatment [25].

## 4. Materials and Methods

### 4.1. Cell Cultures and Reagents

Human NSCLC cell lines, A549 and Lu99 cells, were obtained from American Type Culture Collection, Manassas, and Riken Bioresource Center, Japan, respectively. The cells were grown in RPMI 1640 supplemented with 10% fetal bovine serum (FBS) (Nichirei Biosicence Inc., Tokyo, Japan) [41,51]. Dinactin of ≥95.00% purity (Cat.No. C3454) was purchased from the APExBIO (Boston, MA, USA). Antibodies for Cyclin-A, Cyclin-B, Cyclin-D3, Cdk2, and PCNA (BD Transduction Laboratories™, San Jose, CA, USA), p21 (Santa Cruz Biotechnology, Inc., Dallas, TX, USA), and anti-GAPDH (Trevigen, Minneapolis, MN, USA) were used for the experiments. Metformin was purchased from Sigma-Aldrich (Burlington, MA, USA).

### 4.2. Cell Proliferation and Viability

Cell viability was determined using the 3-(4,5-dimethylthiazol-2-yl)-2,5-diphenyl tetrazolium bromide (MTT) assay [52]. Briefly, one day after seeding 1 × 10^3^ cells/100 µL in 96-well culture plates, the cells were treated with various concentrations of dinactin for an appropriated time. The cells were then treated with 20 μL of 5 mg/mL MTT and dissolved in DMSO (200 μL). Absorbance at 570 nm with a reference at 630 nm were measured by an ELIZA Analyzer ETY-96 (Toyosokki Co. Ltd., Tokyo, Japan). The IC_50_ value was determined by curve fitting with non-linear regression analysis by GraphPad Prism 5.01 (GraphPad Software, Inc., La Jolla, CA, USA).

### 4.3. Cell Cycle Analysis

The cell cycle profile was determined by staining the DNA with propidium iodide and measuring its intensity by flow cytometer [53]. Briefly, Lu99 and A549 cells (3.5 × 10^5^ cells/3.5 mL) were incubated with 0.1–1 µM dinactin, or 24.5 mM metformin as a positive control, for 24 and 48 h. Then, cells were washed with phosphate-buffered saline (PBS) and fixed with ice-cold 100% ethanol for 30 min at 4 °C. After being centrifuged at 3000 rpm for 5 min, the pellets were harvested and treated with RNase A at concentration of 0.25 mg/mL at 37 °C for 30 min. The cells were then stained with 50 µg/mL propidium iodide and incubated at room temperature in the dark for 30 min. After filtration through 50 µm nylon mesh to prevent cell clumping or cell aggregates, the DNA content of 10,000 strained cells in each group was measured with the SA3800 Spectral Cell Analyzer (Sony Biotechnology Inc., San Jose, CA, USA). Data analysis was performed by FlowJo v.10 software (FlowJo, LLC, Ashland, OR, USA).

### 4.4. Western Blot Analysis

Cells were lysed in a strong lysis buffer, as described previously [51]. The lysates were applied to NuPAGE Bis-Tris Gels (Invitrogen, Waltham, MA, USA), and then transferred onto a nitrocellulose membrane. The membrane was then incubated with primary antibody overnight at 8 °C, then incubated with the appropriate horseradish peroxidase-conjugated secondary antibody against rabbit IgG or mouse IgG. Specific bands were then detected with ImmunoStar LD (Wako Pure Chem. Ind. Ltd., Tokyo, Japan) using the C-DiGit Chemiluminescent Western Blot Scanner (LI-COR Bioscience Inc., Lincoln, NE, USA). GAPDH was used as an internal control.

### 4.5. Tumor-Sphere Formation

Lu99 and A549 cells were seeded with or without dinactin at a density of 500 cells/well in serum-free cancer stem cell culture medium (Dulbecco’s modified Eagle’s medium [DMEM]:F12 containing 0.45% methylcellulose, 50 ng/mL epidermal growth factor [EGF], 50 ng/mL fibroblast growth factor [FGF], and B27 supplement) in ultra-low-attachment 96-well plates (Corning Inc., Corning, NY, USA), as previously described [54]. Tumor-spheres will then be solidified into round structures. After 14 days of treatment, tumor-spheres were observed and imaged using an All-in-One microscope (BZ-X700, Keyence, Tokyo, Japan); the number of spheres measuring > 100 µm on a minor axis was counted and their dimensions were recorded.

### 4.6. Quantitative Real-Time RT-PCR (qRT-PCR)

Tumor spheres were separated from parental cells using a filter with a pore size of more than 77 µm (Spheroid Catch, Watson Co. Ltd., Tokyo, Japan) for gene expression analyses, and total RNA was isolated using Isogen (Nippon Gene Co. Ltd., Toyama, Japan). Total RNA was reverse transcribed into cDNA using oligo(dT)16 and MuLV reverse transcriptase (Thermo Fisher Scientific, Waltham, MA, USA), and real-time PCR was conducted using SYBR Green I (LightCycler 480, Roche Lifescience, Basel, Switzerland), as previously described [51,54,55]. The primers used are indicated in Appendix A. GAPDH was used as an internal control, and relative gene expression was calculated as the fold expression.

### 4.7. Statistical Analysis

Values were presented as the mean ± standard deviation (SD) of three independent experiments. The significance of differences between average values of different experimental treatments and controls was assessed by ANOVA, considering that statistical significance was set at a *p* value less than 0.05. When ANOVA revealed significant differences among treatments, post hoc tests were carried out with Dunnett’s Multiple or Bonferroni Comparison Test from GraphPad Prism 5.01

## 5. Conclusions

In this study, we provide a new molecular mechanism for dinatin as an antitumor antibiotic. Our findings indicate the effectiveness of dinactin against lung cancer cells (Lu99 and A549) by inducing G_0_/G_1_ cell cycle arrest through the downregulation of cyclins A, B, D3, and cdk2. Dinactin plays the role of a CSC inhibitor by inhibiting the expression of CSC stemness markers such as *ALDH1A1*, *Nanog*, *Oct4*, and *Sox2* genes. As a consequence, treatment with dinactin suppresses the stemness property associated with cell cycle progression in lung cancer cells. Taken together, these findings suggest that dinactin could be used as an antitumor treatment for NSCLC in the future.

## Figures and Tables

**Figure 1 antibiotics-11-01845-f001:**
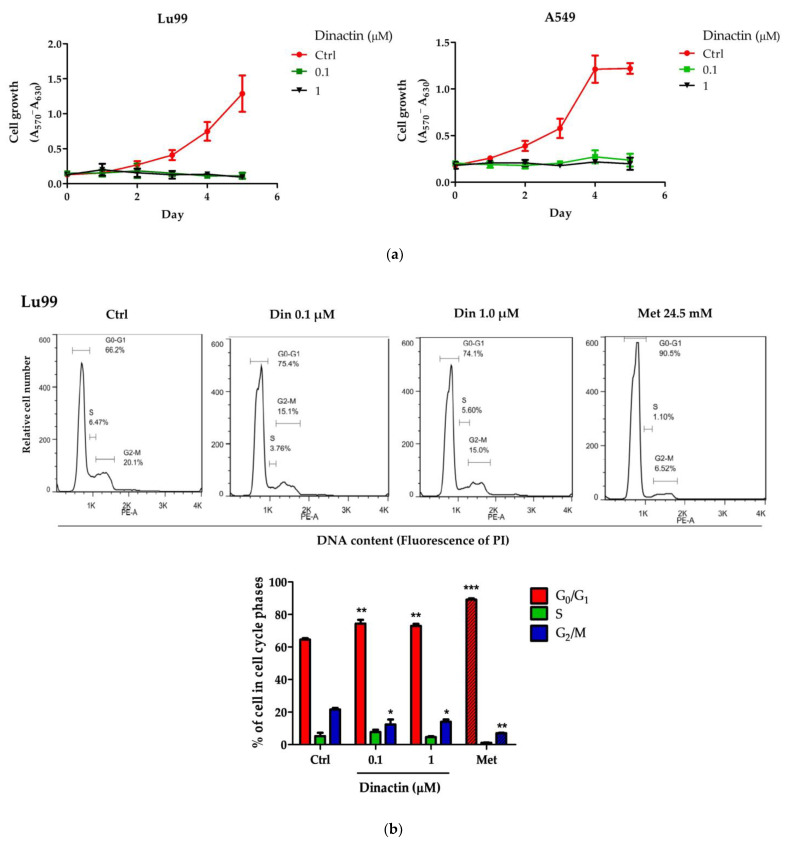
Dinactin inhibited cancer cell growth by inducing G_0_/G_1_ arrest. Lu99 and A549 cells were treated with dinactin 0.1 and 1 µM from 0 to 5 days (**a**). The cell growth was determined by MTT assay. The cell cycle distribution of Lu99 (**b**) and A549 (**c**) cells was analyzed by flow cytometry at 48 h after treatment with 0.1 and 1 μM dinactin. Metformin (24.5 mM) was used as a positive control. The representative histograms show the percentage of cell cycle distribution in the G_0_/G_1_, S, and G_2_/M phases. The data are presented as the mean ± SD of three independent experiments and compared by the two-way ANOVA followed by the Bonferroni post-hoc test. * *p* < 0.05, ** *p* < 0.01, *** *p* < 0.001. Abbreviations: Ctrl, Control; Din, dinactin; Met, Metformin; PI, propidium iodide.

**Figure 2 antibiotics-11-01845-f002:**
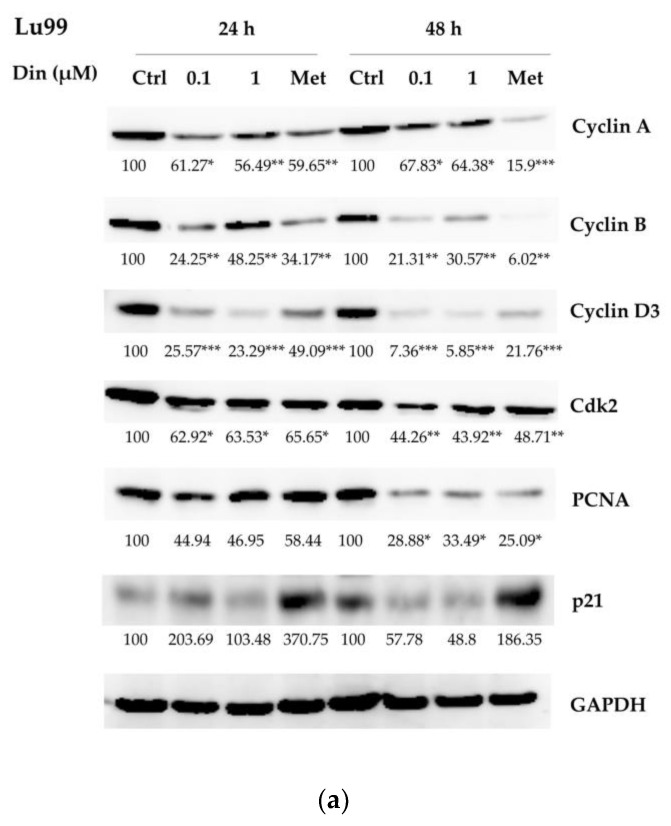
Dinactin down-regulated cell cyclins A, B, and D3, cdk2 and PCNA in Lu99 and A549 cells. Lu99 and A549 cells were treated with 0.1 and 1 μM dinactin for 24 and 48 h. The levels of cell cycle-related proteins were examined by western blot analysis. Representative Lu99 cell results were shown, along with numbers indicating levels of cell cycle-related proteins compared to non-treated cells (expressed as 100), (**a**). GAPDH was used as a loading control. Those of A549 cells were shown (**b**). Metformin (24.5 mM) was used as a loading control and a positive control. The data are presented as the ± SD of three independent experiments and compared by the two-way ANOVA followed by the Bonferroni post-hoc test. * *p* < 0.05, ** *p* < 0.01, *** *p* < 0.001. Abbreviations: Ctrl, Control; Din, dinactin; Met, metformin; Cdk2, cyclin-dependent kinases; PCNA, proliferating cell nuclear antigen; GAPDH, glyceraldehyde 3-phosphate dehydrogenase.

**Figure 3 antibiotics-11-01845-f003:**
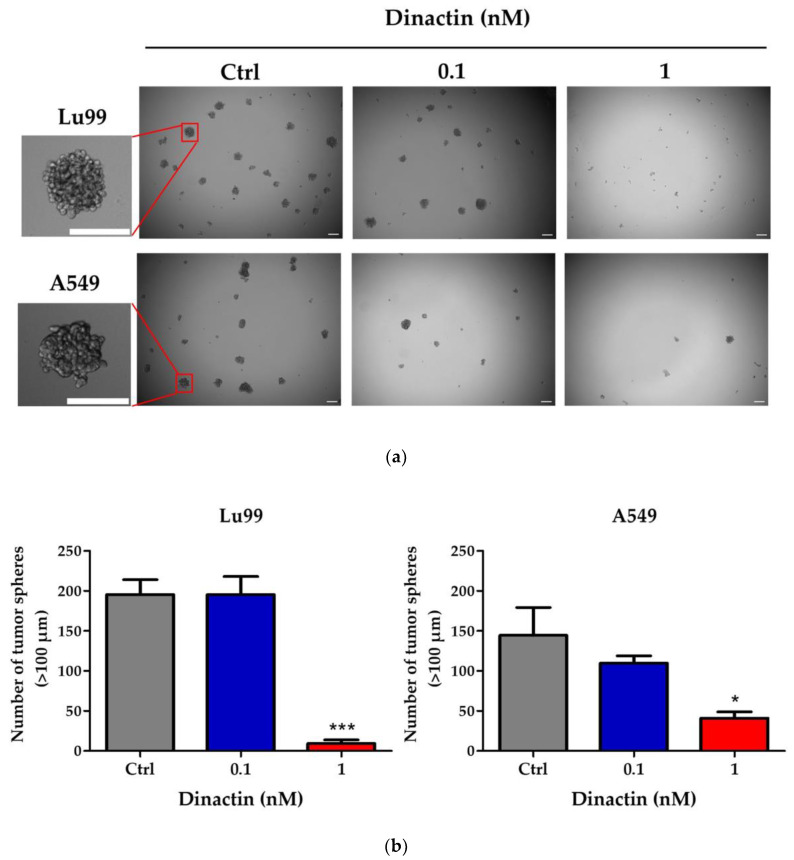
Dinactin inhibited tumor-sphere formation in Lu99 and A549 cells. Lu99 and A549 cells were cultured in serum-free cancer stem cell culture medium in the presence of 0.1 or 1.0 nM of dinactin for 14 days. The representative images of tumor spheres in Lu99 and A549 cells were shown (**a**). White bars indicate 100 µm. The numbers of spheres greater than 100 µm was counted (**b**). The data are presented as the mean ± SD of three independent experiments and compared by the one-way ANOVA followed by Dunnett’s Multiple Comparison Test. * *p* < 0.05, *** *p* < 0.001. Abbreviations: Ctrl, Control.

**Figure 4 antibiotics-11-01845-f004:**
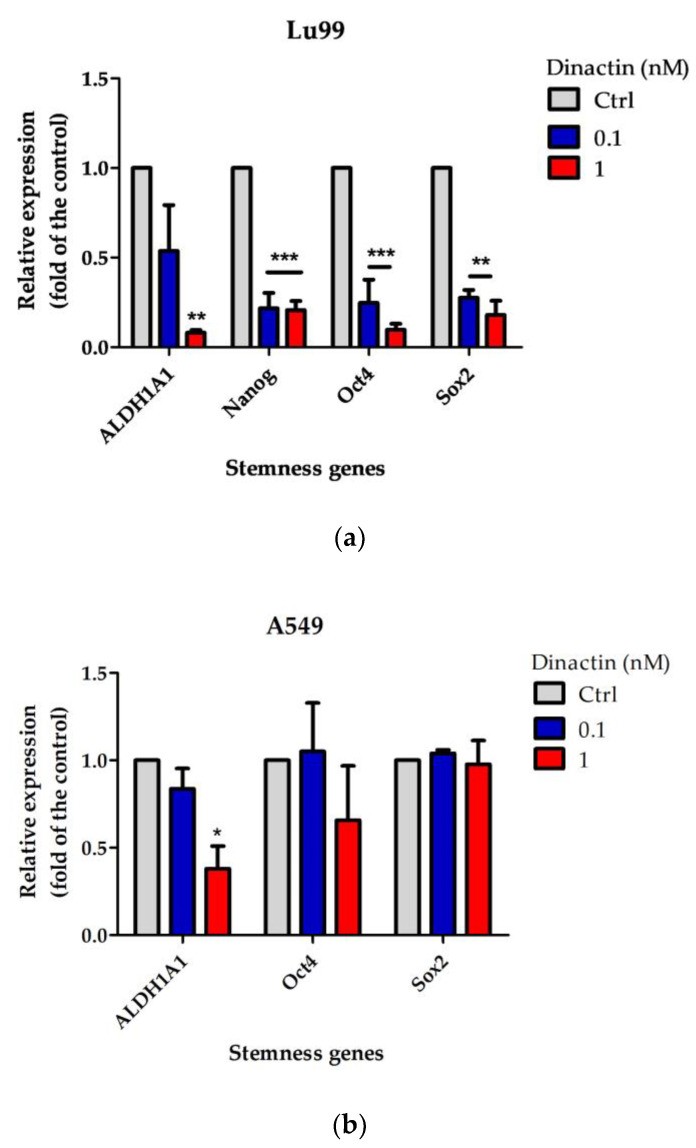
Effects of dinactin on stemness-related gene expressions in Lu99 and A549 tumor- spheres. Lu99 and A549 cells were cultured in serum-free cancer stem cells culture medium in the presence of 0.1 and 1.0 nM of dinactin for 14 days. (**a**) Relative expression of the stemness genes in Lu99 tumor-spheres, and (**b**) those in A549 tumor spheres. Expression levels of the genes were examined by real-time RT-PCR and expressed fold of the control (non-treated tumor spheres). The data are presented as the mean ± SD of three independent experiments and compared by the two-way ANOVA followed by the Bonferroni post-hoc test. * *p* < 0.05, ** *p* < 0.01, *** *p* < 0.001. Abbreviations: Ctrl, Control.

**Table 1 antibiotics-11-01845-t001:** Dinactin reduced the size of tumor-spheres in Lu99 and A549 cells.

Dinactin (nM)	Average Size of Tumor Spheres (µm)	
	Lu99	A549
Control	175.54 ± 55.78	158.72 ± 47.40
0.1	181.04 ± 46.43	159.28 ± 43.98
1	128.82 ± 20.07 ***	134.87 ± 25.71 **

The data are presented as the mean ± SD of all spheres and compared by the one-way ANOVA followed by Dunnett’s Multiple Comparison Test. ** *p* < 0.01, *** *p* < 0.001.

## Data Availability

The data involved in this study are available from the corresponding author upon reasonable request.

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
