# Peer review of "Dinactin: A New Antitumor Antibiotic with Cell Cycle Progression and Cancer Stemness Inhibiting Activities in Lung Cancer"

_antibiotics, 2022, doi:10.3390/antibiotics11121845_

Round 1
Reviewer 1 Report
The authors have shown that antibiotic dinactin can be a candidate to overcome lung carcinoma growth and stemness. However, a few things need to be checked properly.
1. Methods need to be written properly, stemness protocol is not clearly mentioned, and treatment is for three weeks, or spheroids were grown after treatment.
2. Author mentioned finding suggests that the antiproliferative effect of dinactin is not related to the induction of cell death but no evidence suggesting apoptosis assay that either cell is in the apoptotic phase or necrotizing at both doses.
3. Similarly, Although dinactin significantly inhibited tumor-sphere formation in A549 cells, stemness gene expression did not change distinctly. Since Lu99 and A549 cell lines have several differences in malignancy, metastatic potential, and histological types—Lu99 is a lung large cell carcinoma with highly metastatic and A549 is a lung adenocarcinoma—these differences are attributed to differences. The author needs to check the antimetastatic potential of these compounds to support their hypothesis.
4. The cell cycle graph showed especially the control very little S or G2 phase for any specific reason.
5. What is the reason for the filtration of cells using a nylon filter in the cell cycle experiment?
6. What is the correlation between cell cycle and cell stemness.
7. In one cell line there is no significant stemness gene expression as figure 4b showed a very high standard deviation. is it the reason for nonsignificance or might have some issue with the sample preparation for real-time PCR?
Author Response
Responses to reviewer’s comments
(Manuscript ID: antibiotics-2098722)
“Dinactin: A new antitumor antibiotic with cell cycle progression and cancer stemness inhibiting activities in lung cancer”
Please find the responses to the reviewers' comments in the attachment file.
We would like to thank the editor and reviewers for careful and thorough review of this manuscript. We have revised our manuscript in response to your suggestions and the changes have made using “track change.” We hope that this improved manuscript is acceptable for publication in Antibiotics. The answer to their specific comments/suggestions are as follows.
Reviewer #1
The authors have shown that antibiotic dinactin can be a candidate to overcome lung carcinoma growth and stemness. However, a few things need to be checked properly.
Response: Thank you very much for your stimulating comments. According to your suggestion, we completely revised the manuscript, including our careless mistakes. We hope that the revised manuscript will fulfil your request.
- Methods need to be written properly, stemness protocol is not clearly mentioned, and treatment is for three weeks, or spheroids were grown after treatment.
Response: Thank you very much for your valuable suggestions. We edit and add the sentence on lines 367-371.
Original version
Lu99 and A549 cells were seeded at a density of 500 cells/well in serum-free cancer stem cells culture medium (Dulbecco’s modified Eagle’s medium [DMEM]:F12 containing 0.45% methylcellulose, 50 ng/mL epidermal growth factor [EGF], 50 ng/mL fibroblast growth factor [FGF], and B27 supplement) in ultra-low-attachment 96-well plates (Corning Inc., Corning, NY, USA), as previously described [44]. Each treatment was conducted in 24-wells plate. After 3 weeks of culture, all wells were imaged using an All-in-One microscope (BZ-X700, Keyence, Tokyo, Japan), and the number of spheres measuring > 100 µm on a minor axis were counted.
Revised version
Lu99 and A549 cells were seeded at a density of 500 cells/well in serum-free cancer stem cells culture medium (Dulbecco’s modified Eagle’s medium [DMEM]:F12 containing 0.45% methylcellulose, 50 ng/mL epidermal growth factor [EGF], 50 ng/mL fibroblast growth factor [FGF], and B27 supplement) in ultra-low-attachment 96-well plates (Corning Inc., Corning, NY, USA), as previously described [45]. Dinactin concentrations of 0.1 or 1 nM were treated and incubated for 14 days at 37 °C. Tumor-spheres or spheroids will then be formed as solid, round structures. Tumor-spheres were imaged using an All-in-One microscope (BZ-X700, Keyence, Tokyo, Japan), and the number of spheres measuring > 100 µm on a minor axis was counted.
- Author mentioned finding suggests that the antiproliferative effect of dinactin is not related to the induction of cell death but no evidence suggesting apoptosis assay that either cell is in the apoptotic phase or necrotizing at both doses.
Response: Thanks for the stimulating comment. According to Figure 1b on lines 109–121, these results indicated that dinactin is not affected by cell death. Yu et al., 2011 suggested that the accumulation of Sup-G1 indicated cell death, including apoptosis and necrosis. In addition, Kim et al., 2010 and Taniguchi et al., 2008 demonstrated that Sub-G1 indicated cell apoptosis. In order to minimize the bias, we have edited the text so that Sub-G1 indicates "apoptosis" for cell death on line 122 and added more references in the discussion on line 120.
Reference
- Kim, Do-Yeon, Sung-Hak Kang, and Sung-Ho Ghil. "Cirsium japonicum extract induces apoptosis and anti-proliferation in the human breast cancer cell line MCF-7." Molecular Medicine Reports 3, no. 3 (2010): 427-432.
- Taniguchi, Hiroya, Tatsushi Yoshida, Mano Horinaka, Takashi Yasuda, Ahmed E. Goda, Masako Konishi, Miki Wakada, Keisho Kataoka, Toshikazu Yoshikawa, and Toshiyuki Sakai. "Baicalein overcomes tumor necrosis factor–related apoptosis-inducing ligand resistance via two different cell-specific pathways in cancer cells but not in normal cells." Cancer research 68, no. 21 (2008): 8918-8927.
- Yu, Min, Jie Dai, Weiwei Huang, Yang Jiao, Liang Liu, Min Wu, and Deyong Tan. "hMTERF4 knockdown in HeLa cells results in sub-G1 cell accumulation and cell death." Acta Biochim Biophys Sin 43, no. 5 (2011): 372-379.
In addition, we assessed cell morphology by treating cells with 1 µM dinactin for 24, 48, and 60 hrs. The change representing anti-proliferative characteristics, such as cell density, was found to decrease with increasing treatment time, and we did not find the characterization of apoptotic cells. However, in future studies, we need to confirm.
- Similarly, Although dinactin significantly inhibited tumor-sphere formation in A549 cells, stemness gene expression did not change distinctly. Since Lu99 and A549 cell lines have several differences in malignancy, metastatic potential, and histological types—Lu99 is a lung large cell carcinoma with highly metastatic and A549 is a lung adenocarcinoma—these differences are attributed to differences. The author needs to check the antimetastatic potential of these compounds to support their hypothesis.
Response: Thank you for the valuable suggestion. As the reviewer points out, we need to examine the anti-metastatic activity of dinactin by using cell migration assays, cell invasion assays, as well as cell adhesion assays. Therefore, we add the sentence “However, future studies need to investigate the anti-metastatic potential of dinactin.” in the text on line 285-286.
- The cell cycle graph showed especially the control very little S or G2 phase for any specific reason.
Response: Thank you for your stimulating remark. S-phase is the period for DNA synthesis. The transcription involved in the initiation of DNA replication occurs in the G1 phase (Wang et al., 2021). As a result, the percentage of cell population in S phase may be rapidly drifting to G2, resulting in a low population that ranged from 5.20 ±3.61 to 21.60 ± 1.61 in Lu99 cells and 6.83± 3.90 to 15.53 ± 1.22% in A549 cells, respectively. However, the condition of cell culture might impact the cell cycle population even if the protocol was similar. Li et al. 2010, demonstrated that the population of cells in the S phase was 8–15% when A549 was maintained in RPMI-1640 medium supplemented with 10% heat-inactivated fetal bovine serum (FBS), 100000 IU/ml penicillin G, and streptomycin 100 mg/ml (Li et al., 2010). Simultaneously, the S-phase population in A549 cultured in DMEM supplemented with 10% (v/v) heat-inactivated fetal bovine serum (Gibco-BRL) and antibiotics (100 U/ml penicillin and 100 U/ml streptomycin) was 30–60% (Yuan et al., 2014).
References
- Li, Qinglin, Hui Cheng, Guoqi Zhu, Li Yang, An Zhou, Xiaoshan Wang, Nianbai Fang et al. "Gambogenic acid inhibits proliferation of A549 cells through apoptosis-inducing and cell cycle arresting. Biological and pharmaceutical bulletin 33, no. 3 (2010): 415-420.
- Wang, Zhixiang. "Regulation of cell cycle progression by growth factor-induced cell signaling." Cells 10, no. 12 (2021): 3327.
- Yuan, Long, Yongrong Zhang, Juan Xia, Bin Liu, Qingyu Zhang, Jie Liu, Liming Luo, Zhou Peng, Zeqing Song, and Runzhi Zhu. "Resveratrol induces cell cycle arrest via a p53-independent pathway in A549 cells." Molecular medicine reports 11, no. 4 (2015): 2459-2464.
- What is the reason for the filtration of cells using a nylon filter in the cell cycle experiment?
Response: Thank you for the question. The preparation of cells before analysis by flow cytometry has several steps and the possibility of cell clumping or cell aggregates. As a result, we filter the cells with a 50 µm nylon filter to prevent them from aggregating before they enter the machine and detect the single cell. Therefore, we add the sentence on line 350.
Original version
After filtration through 50 µm nylon mesh, DNA content of 10,000 strained cells in each group were measured with SA3800 Spectral Cell Analyzer (Sony Biotechnology Inc, CA, USA).
Revised version
After filtration through 50 µm nylon mesh to prevent cell clumping or cell aggregates, the DNA content of 10,000 strained cells in each group was measured with the SA3800 Spectral Cell Analyzer (Sony Biotechnology Inc, CA, USA).
- What is the correlation between cell cycle and cell stemness.
Response: Thank you for the critical question. We add more discussions in the text on lines 289-303.
“In this study, we focused on ALDH1A1, CD133, NANOG, OCT4, and SOX2, which are transcription factors required for lung cancer pluripotency. Indeed, ALDH1A1 controls cell cycle checkpoints by regulating KLF4 and p21 proteins, which contribute to apoptosis inhibition in cancer stem cells (Meng et al., 2014; Kabakov et al., 2020). CD133 expression levels are linked to cell cycle DNA profiles, and CD133 deficiency reduces cell proliferation significantly. A comparison of wild-type and knockout human embryonic stem cells showed a significantly decreased population in the S phase, whereas the cell population in the G1 phase was increased (Jaksch et al., 2008; Wang et al., 2020). NANOG regulates S-phase entry in human embryonic stem cells via transcriptional regulation of cell cycle regulatory components, resulting in an increase in pluripotency and cell growth (Zhang et al., 2009). OCT4 is a cell cycle promoter that removes the inhibitor p21 of cell cycle progression in the G1 phase and stimulates entry into the S phase, leading to promoted proliferation and cell cycle progression (Lu et al., 2019). SOX2 regulates the cell cycle by interacting with direct and indirect cyclin D as well as cycle inhibitors p21 and p27, resulting in proliferative stem cells (wistowska et al., 2019). Therefore, treatment with dinactin does indeed suppress the stemness property that is related to cell cycle progression in lung cancer cells.”
References
- Kabakov, Alexander, Anna Yakimova, and Olga Matchuk. "Molecular chaperones in cancer stem cells: determinants of stemness and potential targets for antitumor therapy." Cells 9, no. 4 (2020): 892.
- Lu, Yan, Huinan Qu, Da Qi, Wenhong Xu, Shutong Liu, Xiangshu Jin, Peiye Song et al. "OCT4 maintains self-renewal and reverses senescence in human hair follicle mesenchymal stem cells through the downregulation of p21 by DNA methyltransferases." Stem cell research & therapy 10, no. 1 (2019): 1-16.
- Meng, Erhong, Aparna Mitra, Kaushlendra Tripathi, Michael A. Finan, Jennifer Scalici, Steve McClellan, Luciana Madeira da Silva et al. "ALDH1A1 maintains ovarian cancer stem cell-like properties by altered regulation of cell cycle checkpoint and DNA repair network signaling." PloS one 9, no. 9 (2014): e107142.
- Świstowska, Małgorzata, Paulina Gil-Kulik, Arkadiusz Krzyżanowski, Tomasz Bielecki, Marcin Czop, Anna Kwaśniewska, and Janusz Kocki. "Potential effect of SOX2 on the cell cycle of Wharton’s jelly stem cells (WJSCs)." Oxidative Medicine and Cellular Longevity 2019 (2019).
- Zhang, Xin, Irina Neganova, Stefan Przyborski, Chunbo Yang, Michael Cooke, Stuart P. Atkinson, George Anyfantis et al. "A role for NANOG in G1 to S transition in human embryonic stem cells through direct binding of CDK6 and CDC25A." The Journal of cell biology 184, no. 1 (2009): 67-82.
- Marie Jaksch, Jorge Múnera, Ruchi Bajpai, Alexey Terskikh, Robert G. Oshima; Cell Cycle–Dependent Variation of a CD133 Epitope in Human Embryonic Stem Cell, Colon Cancer, and Melanoma Cell Lines. Cancer Res 1 October 2008; 68 (19): 7882–7886.
- Wang, H., Gong, P., Li, J. et al. Role of CD133 in human embryonic stem cell proliferation and teratoma formation. Stem Cell Res Ther 11, 208 (2020). https://doi.org/10.1186/s13287-020-01729-0
- In one cell line there is no significant stemness gene expression as figure 4b showed a very high standard deviation. is it the reason for nonsignificance or might have some issue with the sample preparation for real-time PCR?
Response: Thank you for your inquiry, which brings up an important point. As you realized, in Figure 4b, A549 cells showed a very high standard deviation of ALDH1A1 due to our mistake. We have checked several times and found that dinactin at a concentration of 1 nM reduced the ALDH1A1 gene expression.
Therefore, we edit the result on lines No. 184-185.
Original version
On the other hand, tumor-sphere in A549 cells expressed only three genes (ALDH1A1, Oct4, and Sox2), and dinactin did not show any effects on these genes (Figure 4b).
Revised version
On the other hand, the tumor-sphere in A549 cells expressed only three genes (ALDH1A1, Oct4, and Sox2), and dinactin significantly reduced the relative expression of only ALDH1A1 (Figure 4b).
Reviewer #2
I carefully read the manuscript and found that it is suitable for publication in the prestigious journal. I accept this article for possible publication in response of including more methods of confirmation. There are minor and common mistakes in the article which should be corrected by the authors. After the correction of all the mistakes, the article could be considered for publication in the prestigious antibiotics Journal.
Response: Thank you very much for your favourable and supportive comments. Based on your comments, we revised the manuscript including our mistakes. We hope that these revisions are sufficient for you.
Comments for Authors
Ø The author needs to give the number to each figure on left top side of the image. It is confusing for readers to judge at the middle of the figure. Revised all figures number.
Response: Thank you for bringing this to our attention. Yes, we did.
Ø The author needs to write “Control” or “Ctrl” group instead of Non-treated
Response: Thank you for the valuable suggestion. We agreed to change "Non-treated" to "Ctrl" on all figures and add the abbreviation "Control" to the figure legends.
Ø In tumor sphere formation, the author needs to color it by using crystal violet to check the tumor sphere formation. I suggest to revise the experiment.
Response: Thank you for the reviewer's suggestions. After 14 days of culture, tumor spheres larger than 100 µm were counted and concentrated for RNA extraction for qRT-PCR. Therefore, we are unable to use crystal violet to monitor tumor sphere formation. qRT-PCR also confirmed the presence of CSC markers such as ALDH1A1, Nanog, CD133, Oct4, and Sox2 genes in tumor spheres. As a result, we think revising the experiment is unnecessary. We appreciate your encouragement and understanding.
Ø I suggest that author include the colony formation assay for tumor cells.
Response: Thank you for your favorable suggestions. Yes, such experiment is important. But unfortunately, we have not examined yet. We hope that we will be able to report the data in near future. We are grateful for your encouragement and kind understanding.
Ø The introduction needs to clarify and add the latest references
Response: Thank you for providing such useful information. We agree with the reviewer by adding the sentence “Therefore, exploring the molecular mechanisms of natural bioactive compounds with anti-cancer activity is gaining tremendous interest in the field of oncology [8,9].”and have added the new references No. 8 and 9 on lines 54-56.
- Iqbal, H.; Menaa, F.; Khan, N.U.; Razzaq, A.; Khan, Z.U.; Ullah, K.; Kamal, R.; Sohail, M.; Thiripuranathar, G.; Uzair, B.; et al. Two Promising Anti-Cancer Compounds, 2-Hydroxycinnaldehyde and 2- Benzoyloxycinnamaldehyde: Where do we stand? Comb Chem High Throughput Screen 2022, 25, 808-818, doi:10.2174/1386207324666210216094428.
- Ullah, A.; Ullah, N.; Nawaz, T.; Aziz, T. Molecular mechanisms of Sanguinarine in cancer prevention and treatment. Anti-Cancer Agents in Medicinal Chemistry 2022, 22, doi:10.2174/1871520622666220831124321.
Ø The author needs to include the supplementary file images to the manuscript.
Response: Thank you for your thoughtful suggestion. Yes, we did by including JPG files for supplementary figures S1 and S2.
Ø Mentioned the original dimension clearly in all Figures.
Response: We thank the reviewer for outlining this point. Yes, we did. However, Figure 1 is continued on the next page, so we used "Figure 1 Cont." in the figure legend on page 5/14 as the format of MDPI.
Ø Use EndNote or Mendeley software for references sequences.
Response: Thank you for your thoughtful suggestion. Yes, we did by using the EndNote X9 version.
Ø Check grammatically and spelling throughout the manuscript. There are some mistakes.
Response: Thank you for your kind suggestion. We have proved it through the manuscript. Moreover, this manuscript has been professionally proofread by PRS group. Please find the certificate in the attachment file.
Cite the following references;
v DOI: 10.2174/1871520622666220831124321
v DOI: 10.2174/1386207324666210216094428
Response: Thank you for providing such useful information. We have added these references to Ref. No. 8 and 9 on lines 54-56.
Reviewer #3
Dear authors. The paper submitted for review deals with an important problem of lung cancer treatment. This neoplasm is the most common cause of death from malignant tumors and, in addition, the effectiveness of therapy is very low. The treatment used to date is ineffective and very toxic for the patient.
After reading, I have the following comments:
Response: Thank you very much for your comments and suggestions. Based on your comments, we revised the manuscript and edited the result figure. We hope that these revisions are sufficient for you.
- The lack of results on normal cells in the paper, please supplement or refer to the literature. The assessment of transgenic activity must always be supported by evidence of no toxicity to normal cells and a selective effect of the candidate on the drug.
Response: Thank you for pointing out that this aspect was not clear. The IC50 value of dinactin in normal fibroblast cells was 1.11± mM. So, we added the effect of dinactin on Normal Human Dermal Fibroblast (NHDF) cells in Figure S2 (Line No. 98).
Figure S2. The effect of dinactin on Normal Human Dermal Fibroblast (NHDF) cells.
NHDF cells were treated with various concentrations of dinactin from 0, 0.1, 1, 10, 100, and 1000 nM for 4 days. The IC50 values were determined by curve fitting with non-linear regression analysis (sigmoidal dose response). The values were presented as the mean ±SD of three independent experiments.
2) The microphotographs (Fig. 3) must be larger so that the reader can see something.
Response: Thank you to the reviewer for your valuable suggestions. We agree with the reviewer by making microphotographs on Figure 3a and adding the white bars to indicate 100 µm. We add the sentence “White bars indicate 100 µm.” in the figure legend on line No. 226.
3) Conclusions should be expanded.
Response: Thanking the reviewer for the important suggestion. More sentences are added on lines 391-392, and 395-397.
Original version
Our findings indicate the effectiveness of dinactin against lung cancer cells (Lu99 and A549) by inducing G0/G1 cell cycle arrest through the downregulation of cyclins A, B, and D3, and cdk2. Dinactin plays the role of a CSCs inhibitor by inhibiting the expression of CSC stemness markers such as ALDH1A1, Nanog, Oct4, and Sox2 in Lu99 cells. Taken together, these findings suggest that dinactin could be used as an antitumor treatment for NSCLC in the future.
Revised version
In this study, we provide a new molecular mechanism for dinatin as an antitumor antibiotic. Our findings indicate the effectiveness of dinactin against lung cancer cells (Lu99 and A549) by inducing G0/G1 cell cycle arrest through the downregulation of cyclins A, B, and D3, and cdk2. Dinactin plays the role of a CSCs inhibitor by inhibiting the expression of CSC stemness markers such as ALDH1A1, Nanog, Oct4, and Sox2. As a consequence, treatment with dinactin suppresses the stemness property associated with cell cycle progression in lung cancer cells. Taken together, these findings suggest that dinactin could be used as an antitumor treatment for NSCLC in the future.

Reviewer 2 Report
Journal of antibiotics
Research Article;
The article entitled “Dinactin: A new antitumor antibiotic with cell cycle progression and cancer stemness inhibiting activities in lung cancer’’. Non-small cell lung cancer is one of the most complex diseases, even though there are effective treatments such as chemotherapy and immunotherapy. Since cancer stem cells are responsible for chemo & radio resistance, metastasis, and cancer recurrence, new therapeutic targets for cancer stem cells is precarious. Dinactin produced by microorganisms, has been revealed as an antitumor antibiotic via different mechanisms. The evidence relating to cell cycle progression regulation is reserved, and effects on cancer stemness have not been clarified. This study determined the new function of Dinactin in anti-NSCLC proliferation, focusing on cell cycle progression and cancer stemness properties in Lu99 and A549 cells. Flow cytometry and immunoblotting analyses revealed that Dinactin suppresses cell growth through induction of the G0/G1 phase associated with down-regulation of cyclins A, B, and D3, and cdk2 protein expression. Dinactin reduced both the number and size of the tumor-spheres. PCR analyses showed that Dinactin suppressed sphere formation by significantly reducing expression of CSC markers. Overall, Dinactin could be a promising strategy for NSCLC therapy targeting cancer stem cells.
I carefully read the manuscript and found that it is suitable for publication in the prestigious journal. I accept this article for possible publication in response of including more methods of confirmation. There are minor and common mistakes in the article which should be corrected by the authors. After the correction of all the mistakes, the article could be considered for publication in the prestigious antibiotics Journal.
Comments for Authors
Ø The author needs to give the number to each figure on left top side of the image. It is confusing for readers to judge at the middle of the figure. Revised all figures number.
Ø The author needs to write “Control” or “Ctrl” group instead of Non-treated
Ø In tumor sphere formation, the author needs to color it by using crystal violet to check the tumor sphere formation. I suggest to revise the experiment.
Ø I suggest that author include the colony formation assay for tumor cells.
Ø The introduction needs to clarify and add the latest references
Ø The author needs to include the supplementary file images to the manuscript.
Ø Mentioned the original dimension clearly in all Figures.
Ø Use EndNote or Mendeley software for references sequences.
Ø Check grammatically and spelling throughout the manuscript. There are some mistakes.
Cite the following references;
v DOI: 10.2174/1871520622666220831124321
v DOI: 10.2174/1386207324666210216094428
Author Response

(The authors gave the same response as above.)

Reviewer 3 Report
Dear authors. The paper submitted for review deals with an important problem of lung cancer treatment. This neoplasm is the most common cause of death from malignant tumors and, in addition, the effectiveness of therapy is very low. The treatment used to date is ineffective and very toxic for the patient.
After reading, I have the following comments:
1. The lack of results on normal cells in the paper, please supplement or refer to the literature. The assessment of transgenic activity must always be supported by evidence of no toxicity to normal cells and a selective effect of the candidate on the drug.
2) The microphotographs (Fig. 3) must be larger so that the reader can see something.
3) Conclusions should be expanded.
Author Response

(The authors gave the same response as above.)

Round 2
Reviewer 1 Report
The majority of the points highlighted before were answered, however, still some points need to be cleared.
1. The revised stemness protocol is fine, but it is still not clear that spheroids are treated for 14 days or after 48hrs of treatment spheroids formation was checked for 14 days.
2. In my suggestion there is some evidence required to prove that there is no apoptosis, maybe by doing some apoptotic gene expression instead of FACs assay. (JUST A Suggestion). However, the author proved it by references but not confirmatory.
3. which software used to analyze cell cycle results needs to be mentioned.
Author Response
Responses to reviewer’s comments
Round 2
(Manuscript ID: antibiotics-2098722)
“Dinactin: A new antitumor antibiotic with cell cycle progression and cancer stemness inhibiting activities in lung cancer”
Please find the responses to the reviewers' comments in the attachment file.
We would like to thank the editor and reviewers for careful and thorough review of this manuscript. We have revised our manuscript in response to your suggestions and the changes have made using “track change.” The edits from Round 2 were labeled in green. We hope that this improved manuscript is acceptable for publication in Antibiotics. The answer to their specific comments/suggestions are as follows.
Reviewer #1
The majority of the points highlighted before were answered, however, still some points need to be cleared.
Response: Thank you very much for your stimulating comments that helped complete this manuscript. We completely revised the manuscript in response to your suggestions. We hope that the revised manuscript will meet your requirements.
- The revised stemness protocol is fine, but it is still not clear that spheroids are treated for 14 days or after 48hrs of treatment spheroids formation was checked for 14 days.
Response: Thank you very much for your valuable suggestions. We apologize for any confusion caused by the spheroid formation assay explanation. Cells were seeded with or without dinactin at a concentration of 0.1 or 1 nM. Spheroids were treated for 14 days, then observed and counted. And we found that dinactin treatment reduced the size and number of spheroids.
Therefore, we edit and add the sentence on lines 367-371.
Original version
Lu99 and A549 cells were seeded at a density of 500 cells/well in serum-free cancer stem cells culture medium (Dulbecco’s modified Eagle’s medium [DMEM]:F12 containing 0.45% methylcellulose, 50 ng/mL epidermal growth factor [EGF], 50 ng/mL fibroblast growth factor [FGF], and B27 supplement) in ultra-low-attachment 96-well plates (Corning Inc., Corning, NY, USA), as previously described [44]. Each treatment was conducted in 24-wells plate. After 3 weeks of culture, all wells were imaged using an All-in-One microscope (BZ-X700, Keyence, Tokyo, Japan), and the number of spheres measuring > 100 µm on a minor axis were counted.
Revised version
Lu99 and A549 cells were seeded at a density of 500 cells/well in serum-free cancer stem cells culture medium (Dulbecco’s modified Eagle’s medium [DMEM]:F12 containing 0.45% methylcellulose, 50 ng/mL epidermal growth factor [EGF], 50 ng/mL fibroblast growth factor [FGF], and B27 supplement) in ultra-low-attachment 96-well plates (Corning Inc., Corning, NY, USA), as previously described [45]. Dinactin concentrations of 0.1 or 1 nM were treated and incubated for 14 days at 37 °C. Tumor-spheres or spheroids will then be formed as solid, round structures. Tumor-spheres were imaged using an All-in-One microscope (BZ-X700, Keyence, Tokyo, Japan), and the number of spheres measuring > 100 µm on a minor axis was counted.
Revised version Round 2
Lu99 and A549 cells were seeded with or without dinactin at a density of 500 cells/well in serum-free cancer stem cell culture medium (Dulbecco’s modified Eagle’s medium [DMEM]:F12 containing 0.45% methylcellulose, 50 ng/mL epidermal growth factor [EGF], 50 ng/mL fibroblast growth factor [FGF], and B27 supplement) in ultra-low-attachment 96-well plates (Corning Inc., Corning, NY, USA), as previously described [45]. Tumor-spheres will then be solidified into round structures. After 14 days of treatment, tumor-spheres were observed and imaged using an All-in-One microscope (BZ-X700, Keyence, Tokyo, Japan), and the number of spheres measuring > 100 µm on a minor axis was counted, and their dimensions were recorded.
- In my suggestion there is some evidence required to prove that there is no apoptosis, maybe by doing some apoptotic gene expression instead of FACs assay. (JUST A Suggestion). However, the author proved it by references but not confirmatory.
Response: Thanks again for the stimulating comment that will lead us to re-thinking. Yes, we agree with the reviewer that apoptosis confirmation is actually needed. In the future, it will be necessary to determine the expression of apoptosis genes such as p53, bak, bax, bcl-2, bcl-x, caspase-3, and caspase-9.
Therefore, we edit the sentence on lines 117–118.
Original version
It is important to note that the sub-G1 peak, an indicator of apoptotic cell death, was not observed in both cells. These findings suggest that the anti-proliferative effect of dinactin in Lu99 and A549 cells is due to cell cycle arrest at the G0/G1 phase rather than cell dead induction.
Revised version
It is important to note that the sub-G1 peak, an indicator of apoptotic cell death, was not observed in both cells. These findings suggest that the anti-proliferative effect of dinactin in Lu99 and A549 cells is due to cell cycle arrest at the G0/G1 phase rather than apoptosis induction.
Revised version Round 2
These results suggest that dinactin suppresses the cell growth of Lu99 and A549 through induction of the G0/G1 phase of cell cycle arrest.
In addition, we edit the sentence on lines 132-134.
Original version
These results coincided with the results that dinactin suppresses cell growth by induction of G0/G1 phase cell cycle arrest rather than apoptosis. Dinactin induced G0/G1 arrest by the downregulation of cyclins A, B, and D3, cdk2, and PCNA, resulting in inhibition of cancer cell proliferation.
Revised version Round 2
These results support the finding that dinactin suppresses cell growth by inducing G0/G1 phase cell cycle arrest via the downregulation of cyclins A, B, and D3, cdk2, and PCNA, resulting in inhibition of cancer cell proliferation.
- which software used to analyze cell cycle results needs to be mentioned.
Response: Thank you for the reviewer's suggestion. The data were analyzed by the Flowjo version 10 software. So, we add the sentences “Data analysis was performed by FlowJo v.10 software (FlowJo, LLC, Ashland, OR, USA).” on line 349-350.
